# Emotion Recognition Using Different Sensors, Emotion Models, Methods and Datasets: A Comprehensive Review

**DOI:** 10.3390/s23052455

**Published:** 2023-02-23

**Authors:** Yujian Cai, Xingguang Li, Jinsong Li

**Affiliations:** School of Electronic Information Engineering, Changchun University of Science and Technology, Changchun 130022, China

**Keywords:** sensors for emotion recognition, emotion models, emotional signal processing, classifiers, emotion recognition datasets

## Abstract

In recent years, the rapid development of sensors and information technology has made it possible for machines to recognize and analyze human emotions. Emotion recognition is an important research direction in various fields. Human emotions have many manifestations. Therefore, emotion recognition can be realized by analyzing facial expressions, speech, behavior, or physiological signals. These signals are collected by different sensors. Correct recognition of human emotions can promote the development of affective computing. Most existing emotion recognition surveys only focus on a single sensor. Therefore, it is more important to compare different sensors or unimodality and multimodality. In this survey, we collect and review more than 200 papers on emotion recognition by literature research methods. We categorize these papers according to different innovations. These articles mainly focus on the methods and datasets used for emotion recognition with different sensors. This survey also provides application examples and developments in emotion recognition. Furthermore, this survey compares the advantages and disadvantages of different sensors for emotion recognition. The proposed survey can help researchers gain a better understanding of existing emotion recognition systems, thus facilitating the selection of suitable sensors, algorithms, and datasets.

## 1. Introduction

Emotion is a comprehensive manifestation of people’s physiological and psychological states; emotion recognition was systematically proposed in the 1990s [1]. With the rapid development of science and technology, emotion recognition has been widely used in various fields, such as human-computer interactions (HCI) [2], medical health [3], Internet education [4], security monitoring [5], intelligent cockpit [6], psychological analysis [7], and the entertainment industry [8].

Emotion recognition can be realized through different detection methods and different sensors. Sensors are combined with advanced algorithm models and rich data to form human-computer interaction systems [9,10] or robot systems [11]. In the field of medical and health care [12], emotion recognition can be used to detect the patient’s psychological state or adjuvant treatment, and improve medical efficiency and medical experience. In the field of Internet education [13], emotion recognition can be used to detect students’ learning status and knowledge acceptance, and cooperate with relevant reminders to improve learning efficiency. In the field of criminal interrogation [14], emotion recognition can be used to detect lies (authenticity test). In the field of intelligent cockpits [15], it can be used to detect the drowsiness and mental state of the driver to improve driving safety. In the field of psychoanalysis [16], it can be used to help analyze whether a person has autism. This technique can also be applied to recognize the emotions of the elderly, infants, and those with special diseases who cannot clearly express their emotions [17,18].

A correct understanding of emotion can deepen the research on emotion recognition. Section 2 introduces the definition of emotion and famous emotion models. Each sensor has different detection emphases for emotion recognition, which can be roughly divided into three categories: The first one is to use human actions or speech signals, such as facial expressions, speech, and gestures. The second one is to use the physiological signals inside the human body, such as EEG, respiratory rate, and heart rate. The last type is multi-modal fusion emotion recognition, which uses multiple signals for emotion recognition. These three types of detection methods have their own advantages and disadvantages, which are detailed in Section 3. Preprocessing, feature extraction, and classification methods for different sensor signals are detailed in Section 4 and Section 5. Section 6 presents some of the main datasets for different signals. Section 7 and Section 8 are the conclusion of the overall survey and thoughts on the future development of emotion recognition.

## 2. Emotion Models

The definition of emotion is the basis of emotion recognition. The basic concept of emotion was proposed by Ekman in the 1970s [19]. At present, there are two mainstream emotion models: the Discrete emotion model and the dimensional emotion model.

### Discrete Emotion Model

Darwinian evolution [20] holds that emotions are primitive or fundamental. Emotion as a form is considered to correspond to discrete and elementary responses or tendencies of action. The discrete emotion model divides human emotions into limited categories [21], mainly including happiness, sadness, fear, anger, disgust, surprise, etc. There are two to eight basic emotions, according to different theories. However, these discrete emotion model theories have certain common features. They believe that emotions are: mental and physiological processes; caused by the awareness of developmental events; inducing factors for changes in the body’s internal and external signals; related to a fixed set of actions or tendencies. Ekman proposed seven characteristics to distinguish different basic emotions and emotional phenomena: autonomous evaluation; have specific antecedent events; also present in other primates; rapid onset; short duration; unconscious or involuntary appearance; reflected in unique physiological systems such as the nervous system and facial expressions. R. Plutchik proposed eight basic emotions and distinguished them according to intensity, forming the Plutchik’s wheel model [22]. It is a well-known discrete emotion model, as shown in Figure 1 (adapted from [22]).

Dimensional emotion models view emotions as combinations of vectors within a more fundamental dimensional space. This enables complex emotions to be researched and measured in fewer dimensions. Core emotions are generally expressed in two-dimensional or three-dimensional space. The dimensional emotion model in the two-dimensional space is usually the arousal-valence model. Valence reflects the positive or negative evaluation of an emotion and the degree of pleasure the participant feels. Arousal reflects the intensity or activation of an emotion in the body. The level of arousal reflects the individual’s will, and low arousal means less energy. However, dimensional emotion models in two dimensions were not able to successfully distinguish core emotions with the same degree of consistency and valence. For example, both anger and fear have high arousal and low valence. Therefore, a new dimension needs to be introduced to distinguish these emotions.

The most famous three-dimensional emotion model is the pleasure, arousal, and dominance (PAD) model [23] proposed by Mehrabian and Russell through the study of environmental psychology methods [24] and the feeling-thinking-acting [25] model, as shown in Figure 2 (adapted from [23]).

Dominance represents control or position, and indicates whether a certain emotion is submissive. It is worth noting the dimensional emotion model can accurately identify the core emotion. However, for some complex emotions, the dimensional emotion model will lose some details.

## 3. Sensors for Emotion Recognition

The sensors used for emotion recognition mainly include visual sensors, audio sensors, radar sensors, and other physiological signal sensors, which can collect signals of different dimensions and achieve emotional analysis through some algorithms. Different sensors have different applications in emotion recognition. The advantages and disadvantages of different sensors for emotion recognition are shown in Table 1.

### 3.1. Visual Sensor

Emotion recognition based on visual sensors is one of the most common emotion recognition methods. It has the advantages of low cost and simple data collection. At present, visual sensors are mainly used for facial expression recognition (FER) [28,29,30] to detect emotion or remote photoplethysmography (rPPG) technology to detect heart rate [31,32]. The accuracies of these methods severely drop as the light intensity decreases.

The facial expression recognition process is shown in Figure 3. Facial expressions can intuitively reflect people’s emotions. It is difficult for machines to capture the details of expressions like humans [33]. Facial expressions are easy to hide, which leads to emotion recognition errors [34]. For example, in some social activities, we usually politely smile even though we are not in a happy mood [35].

Different individuals have different skin colors, looks, and facial features [36,37], which pose challenges to the accuracy of classification. Facial features of the same emotion can be different, and small changes in different emotions of the same individual are not very obvious [38]. Therefore, there is a classification challenge of large intra-class distance and small inter-class distance for emotion detection through facial expression recognition by the camera. It is also difficult to effectively recognize emotions when the face is occluded (wearing a mask) or from different shooting angles [39].

Photoplethysmography (PPG) is an optical technology for the non-invasive detection of various vital signs, which was first proposed in the 1930s [40]. PPG is widely used in the detection of physiological signals in personal portable devices (smart wristbands, smart watches, etc.) [41,42]. The successful application of PPG has led to the rapid development of remote photoplethysmography (rPPG). A multi-wavelength RGB camera is used by rPPG technology to identify minute variations in skin color on the human face caused by changes in blood volume during a heartbeat [43], as shown in Figure 4.

The rPPG technology can be used to obtain the degree of peripheral vascular constriction and analyze the participant’s emotions. External vasoconstriction is considered to be a defensive physiological response. When people are in a state of pain, hunger, fear, or anger, the constriction of external blood vessels will be enhanced. Conversely, in a calm or relaxed state, this response will reduce.

With the improvement in hardware and algorithm level, rPPG technology can also realize remote non-contact monitoring and estimation of heart rate [44], respiratory rate [45], blood pressure [46], or other signals. Emotion recognition is performed after analyzing a large amount of monitoring data. These signals can classify emotions into a few types and intensities. There are certain errors in the recognition of multiple types of emotions. It is necessary to combine other physiological information to improve the accuracy rate of emotion recognition [47].

### 3.2. Audio Sensor

Language is one of the most important components of human culture. People can express themselves or communicate with others through language. Speech recognition [48] has promoted the development of speech emotion recognition (SER) [49]. Human speech contains rich information that can be used for emotion recognition [50,51]. Understanding the emotion in information is essential for artificial intelligence to engage in effective dialog. SER can be used for call center dialog, automatic response systems, autism diagnosis, etc. [52,53,54]. SER is jointly completed by acoustics feature extraction [55] and language mark [56]. The process of SER is shown in Figure 5.

In the preprocessing stage, the input signal is enhanced into segmentations [57] after noise reduction; and then feature extraction and classification are performed [58]. The language model [59] can identify emotional expressions with specific semantic contributions. The acoustic model can distinguish different emotions contained in the same sentence by analyzing the features of prosody or spectral [60]. Combining these two models can improve the accuracy of SER.

Understanding the emotion in speech is a complex process. Different speaking styles of different people will bring about acoustic variability, which will directly affect speech feature labeling and extraction [61]. The same sentence may contain different emotions [62], and some specific emotional differences often depend on the speaker’s local culture or living environment, which also pose challenges for SER.

### 3.3. Radar Sensor

Different emotions will cause a series of physiological responses, such as changes in respiratory rate [63], heart rate [64,65], brain wave [66], blood pressure [67], etc. For example, the excitement caused by happiness, anger, or anxiety can lead to an increased heart rate [68]. Positive emotions can increase respiratory rate, and depressive emotions can tend to inhibit breathing [69]. Respiratory rate also affects heart rate variability (HRV), which decreases when exhaling and increases when inhaling [70]. Currently, radar technology is widely used in remote vital signs detection [71] and wireless sensing [72]. Radar sensors can use the echo signal of the target to analyze the chest micro-motion caused by breathing and heartbeats. It can realize remote acquisition of these physiological signals. The overall process of emotion recognition based on radar sensor is shown in Figure 6.

Compared with visual sensors, emotion recognition based on radar sensors is unrestricted by light intensity [73]. However, in real environments, radar echo signals are affected by noise, especially for the radial doppler motion close to or away from the radar [74], which affects the accuracy of sentiment analysis.

### 3.4. Other Physiological Sensors

Emotions have been shown to be biological since ancient times. Excessive emotion is believed to have some effects on the functioning of vital organs. Aristotle believed that the influence of emotions on physiology is reflected in the changes in physiological states, such as a rapid heartbeat, body heat, or loss of appetite. William James first proposed the theory of the physiology of emotion [75]. He believed that external stimuli would trigger activity in the autonomic nervous system and create a physiological response in the brain. For example, when we feel happy, we laugh; when we feel scared, our hairs stand on end; when we feel sad, we cry.

Human emotion is a spontaneous mental state, which is reflected in the physiological changes of the human body and significantly affects our consciousness [76]. Many other physiological signals in the human body, such as electroencephalogram (EEG) [66], electrocardiogram (ECG) [77], electromyogram (EMG) [78], galvanic skin response (GSR) [79], blood volume pulse (BVP) [80], and electrooculography (EOG) [81], as shown in Figure 7.

EEG measures the electrical signal activity of the brain by setting electrodes on the skin surface of the head. Many studies have shown that the prefrontal cortex, temporal lobe, and anterior cingulate gyrus of the brain are related to the control of emotions. Their levels of activity induce emotions such as anxiety, irritability, depression, worry, and resentment, respectively.

ECG is a method of electrical monitoring on the surface of the skin that detects the heartbeat controlled by the body’s electrical signals. Heart rate and heart rate variability obtained through subsequent analysis are widely used in affective computing. Heart rate and heart rate variability are controlled by the sympathetic nervous systems and parasympathetic nervous systems. The sympathetic nervous system can speed up the heart rate, which is reflected in greater psychological stress and activation. The parasympathetic nervous system is responsible for bringing the heart rate down to normal levels, putting the body in a more relaxed state.

EMG measures the degree of muscle activation by collecting the voltage difference generated during muscle contraction. The current EMG signal measurement technology can be divided into two types. The first is to study facial expressions by measuring facial muscles. The second is to place electrodes on the body to recognize emotional movements.

GSR is another signal commonly used for emotion recognition. Human skin is normally an insulator. When sweat glands secrete sweat, the electrical conductivity of the skin will change Therefore, GSR can reflect the sweating situation of a person. GSR is usually measured on the palms or soles of the feet, where sweat glands are thought to best reflect changes in emotion. When a person is in an anxious or tense mood, the sweat glands usually secrete more sweat, which causes a greater change in current.

Related physiological signals also include BVP, EOG, etc. These signals all change with emotional changes, and they are not subject to human conscious control [82]. Therefore, these signals can be measured by different physiological sensors to achieve the purpose of emotion recognition. Using these physiological sensors can accurately and quickly obtain real human physiological signals. However, physiological sensors other than visual, audio, and radar sensors usually need to touch the skin or wear related equipment to extract physiological signals, which will affect people’s daily comfort (most people will not accept this monitoring method). Contact sensors are limited by weight and size [27]. These contact devices may also cause people tension and anxiety, which will affect the accuracy of emotion recognition.

### 3.5. Multi-Sensor Fusion

There are certain deficiencies in single-modal emotion recognition, and it is usually unable to accurately identify complex emotions. The multi-modal emotion recognition method refers to the use of signals obtained by multiple sensors to complement each other and obtain more reliable recognition results. Multi-modal approaches can promote the development of emotion recognition. Multi-modal emotion recognition can often achieve the best recognition performance, but the computational complexity will increase due to the excessive number of channels. There are higher requirements for the collection of multi-modal datasets. Multi-modal emotion recognition has different fusion strategies, which can be mainly divided into pixel-level fusion, feature-level fusion, and decision-level fusion.

Pixel-level fusion [83] refers to the direct fusion of the original data; the semantic information and noise of the signal will be superimposed, which will affect the classification effect after fusion. Processing time is wasted when there is too much redundant information.

The feature-level fusion [84] process is shown in Figure 8. Feature-level fusion occurs in the early stages of the fusion process. Extract features from different input signals and combine them into high-dimensional feature vectors. Finally, output the result through a classifier. Feature-level fusion retains most of the important information, it can greatly reduce computing consumption. However, when the amount of data is small or some details are missing, the final accuracy rate will decrease.

The decision-level fusion [85,86] process is shown in Figure 9. Decision-level fusion refers to the fusion of independent decisions of each part after making independent decisions based on signals collected by different sensors.

The advantage of decision-level fusion is that independent feature extraction and classification methods can be set according to different signals. It has lower requirements for the integrity of multimodal data. Decision-level fusion has higher robustness and better classification results.

## 4. Emotion Recognition Method

Choosing the right method can improve the accuracy of emotion recognition [87]. The emotion recognition method of different sensors is described in Figure 10.

Signal preprocessing refers to improving signal quality and reducing noise. Feature extraction is mainly used to find the characteristics of different signals and reduce the amount of calculation required for classification. Classification refers to applying the extracted features to a certain classification model. Finally, the emotion corresponding to the signal is obtained through analysis.

### 4.1. Signal Preprocessing

For emotion recognition from different sensors, signal preprocessing is an important step [88]. Preprocessing can reduce the impact of noise in the early stages of emotion recognition.

For visual signals, mainly use cropping, rotation, scaling, grayscale, and other methods for signal preprocessing.

The signal preprocessing method for audio signals mainly includes:Silent frame removal: Remove frames below the set threshold to reduce calculation consumption [89];Pre-emphasis: Compensate for high-frequency components of the signal;Regularization: Adjust the signal to a standard level to reduce the influence of different environments on the results;Window: Prevent signal edge leakage from affecting feature extraction [90];Noise reduction algorithm: Use noise reduction algorithms such as minimum mean square error (MMSE) to reduce background noise.

The preprocessing methods for radar signals and physiological signals mainly include:Filtering: Remove noise, signal crosstalk [91], or baseline wander [92] by different filters;Wavelet transform [93]: Using time window and frequency window to characterize the local characteristics of physiological signals;Nonlinear dynamics: Use approximate entropy [94], sample entropy [95], transfer entropy [96] to obtain a smooth signal estimate and remove transient disturbances [97].

### 4.2. Feature Extraction

Feature extraction can ignore information irrelevant to the target, reduce the amount of calculation, overcome the curse of dimensionality, and improve the generalization ability of the model. Signals often require feature extraction before being input into some classical classification models.

#### 4.2.1. For Visual Signals

**Principal Component Analysis (PCA)** [98]: PCA is a very common dimensionality reduction method. Keeping the most important features of high-dimensional data while removing noise and unimportant features. This can greatly improve data processing speed and save time and costs. PCA can be defined as an orthogonal linear transformation that projects data to a new coordinate system. PCA can satisfy the maximum reconfiguration, which means that the distance between the sample point and the hyperplane is close enough. At the same time, PCA can also satisfy maximum separability, which means that the projection of sample points onto the hyperplane can be separated as much as possible.

In [99], the authors simply used PCA to reduce the dimensionality of the feature vectors. The accuracy rate on the JAFFE dataset reached 74.14%. In [100], the authors combined PCA and PSO to obtain optimized feature vectors. The accuracy rate on the JAFFE dataset reached 94.97%. In [101], the authors proposed two-dimensional PCA; 2DPCA is based on 2D image matrices instead of 1D vectors, so there is no need to convert image matrices to vectors before feature extraction. Indeed, 2DPCA can directly use the original image to construct the covariance matrix, which is more effective than PCA. In [102], the authors utilized bidirectional PCA to extract visual features. The accuracy rate on the YALE multimodal dataset reached 94.01, which was an increase of 0.9% compared with the PCA method.

**Histogram of Oriented Gradients (HOG)** [103]: HOG was proposed based on image edge information and was first used for object detection. Each window region of an image can be described by the local distribution and gradient of edge directions. A HOG descriptor can be obtained by computing the histogram of edge directions in these cells and normalizing them. Combining these descriptors can be used to detect facial expressions. The features generated by HOG are not affected by illumination and geometric transformation.

In [104], the authors proposed a framework for emotion recognition based on HOG and SVM. The accuracy rate on the GEMEP-FERA dataset reached 70%. In [105], the authors proposed a FER framework for real-time inference of emotional states. The framework extracted HOG features from active face patches; 95% accuracy was achieved on the CK+ dataset. In [106], the authors proposed an emotion recognition framework based on HOG descriptors and the Cuttlefish algorithm. This method did not generate irrelevant or noisy features. The model achieved 97.86%, 95.15%, and 90.95% accuracy on the CK+, RaFD, and JAFFE datasets.

Other feature extraction methods for visual signals include Local Binary Patterns (LBP) [107] and Linear Discriminate Analysis (LDA) [108].

#### 4.2.2. For Speech Signals

Speech signal features mainly include prosodic features [109], frequency spectral features, frequency cepstral coefficients [110], and energy features. These features carry both information and emotion. Therefore, some methods can be utilized to extract them.

**Linear Predictor Coefficients (LPC)** [111]: LPC is based on a speech production model. This model uses an all-pole filter to model the characteristics of the vocal tract. LPC is equivalent to the smooth envelope of the speech logarithmic spectrum. It can be directly computed from windowed parts of speech by autocorrelation or covariance methods. LPC can accurately and quickly estimate speech parameters.

In [112], the authors combined the features of TEO and LPC for T-LPC feature extraction. This method can accurately recognize stress speech signals. The accuracy on the Emo-DB dataset reached 82.7% (male) and 88% (female). In [113], the authors proposed a combined spectral coefficient optimization method based on LPC. The accuracy on the Emo-DB dataset reached 88%. Comparative experiments showed that this optimization method improved the accuracy by 4%. In [114], the authors measured the emotion recognition accuracy when LPC coefficients were introduced in the feature vectors. Using only the LOC coefficients, the model achieved 78% accuracy on the SROL dataset. In [115], the authors proposed a meta-heuristic feature selection model. This model took LPC features as input. The accuracy of the model on the SAVEE and Emo-DB datasets reached 97.31% and 98.46%.

**Teager Energy Operator(TEO)** [116]: TEO is a powerful nonlinear energy operator. It is able to extract signal energy based on mechanical and physical considerations. TEO can extract the features of speech when the utterance presents a certain stress. It measures speech non-proximity by processing the characteristics of speech signals in the frequency and time domains.

In [117], the authors proposed a two-stage emotion recognition system based on TEO. Autoencoders improved recognition rates. The accuracy on the RML dataset reached 74.07%. In [118], the authors proposed the EMD-TEO model. Experiments showed that the features extracted based on TEO were robust, and the performance of speech emotion recognition was significantly improved. The accuracy of the model on the EMO-DB dataset reached 81.34%. In [119], the authors fused TEO and MFCC to form T-MFCC feature extraction technology. TEO extracted the nonlinear features of speech and was mainly used to identify stressful emotions. Experiments showed that T-MFCC had better performance. The accuracy of the model on the EMO-DB dataset reached 93.33%.

Other commonly used speech signal feature extraction methods include Short-time Coherence (SMC), Fast Fourier Transform (FFT), Principal Component Analysis (PCA) [120], and linear discriminant analysis (LDA) [121].

#### 4.2.3. For Physiological and Radar signals

**Fast Fourier Transform (FFT)** [122]: FFT is a popular signal processing method. It can be used to convert time-domain signals to frequency-domain signals, and vice versa. For spectrum analysis, the magnitude squared of the FFT is usually used to obtain the Power Spectral Density (PSD). PSD can be used to analyze the contribution of a specific frequency band to the total power of the signal.

In [123], the authors utilized FFT to analyze short-duration EEG signals for emotion classification. Through experiments, it was concluded that the short-term EEG signal characteristics reflected the changes in emotional state. The accuracy on the self-built dataset reached 91.33%. In [124], the authors built an emotion recognition model based on FFT and Genetic Programming (GP). FFT was used to convert a signal from the time domain to the frequency domain. The accuracy on the self-built dataset (four emotions) reached 89.14%. In [125], the authors utilized FFT and Wigner-Ville Distribution (WVD) methods to convert physiological signals into images. Putting the image into a CNN model could obtain excellent classification results. The accuracy on the self-built dataset reached 93.01%.

**Maximal-Relevance Minimal-Redundancy (mRMR)** [126]: mRMR uses mutual information as a correlation measurement with maximum dependence criterion and minimum redundancy criterion. It is capable of selecting features with the strongest correlation with the categorical variable. The mRMR algorithm can not only reduce dimensions and improve prediction accuracy, but also obtain features with more meaning and value.

In [127], the authors researched stable patterns over time for emotion recognition from EEG. The model used the mRMR algorithm to reduce the dimension and improve the stability of the classifier. The accuracy on the DEAP dataset and SEED dataset reached 69.67% and 91.07%. In [128], the authors proposed a method that combined the feature selection task of mRMR and kernel classifiers for emotion recognition. The authors used mRMR to incorporate feature selection tasks into classification tasks. The accuracy on the DEAP dataset reached 60.7% (Arousal) and 62.33% (Valence). In [129], the authors analyzed non-stationary physiological signals and extracted features that could be used to achieve accurate emotion recognition. The authors utilized the mRMR algorithm to reduce the dimensionality of the constructed feature vectors. The average accuracy on the DEAP dataset reached 80%.

Other feature extraction methods of physiological signals and radar signals include Empirical Mode Decomposition (EMD), Linear Discriminate Analysis (LDA) [130], Locality Preserving Projections (LPP) [131], and the Relief-F algorithm [132].

## 5. Classification

The classifier can classify different input signals and output the corresponding emotion category. The quality of the classifier will affect the accuracy of emotion recognition. The current classification methods can be divided into two categories: Classical machine learning methods and deep learning methods. This section will introduce several commonly used machine learning methods and deep learning methods.

### 5.1. Machine Learning Methods

#### 5.1.1. SVM

Support vector machine (SVM) [133] aims to find the hyperplane with the largest interval in the sample space to produce more robust classification results, as shown in Figure 11. For more complex samples, it can be mapped from the original space to a higher dimensional space. Solving the corresponding kernel function [134] makes these samples linearly separable in the feature space. Soft margins [135] and regularization can be added to prevent overfitting of the trained model.

The hyperplane can be described by $w^{T}x+b$. $w=\left(w_{1};w_{2};\ldots ;w_{d}\right)$ is the normal vector, which determines the direction of the hyperplane. b is the displacement term, which determines the distance between the hyperplane and the origin. The distance from any point x in the sample space to the hyperplane $\left(w,b\right)$ is
(1)$$r=\frac{\left|w^{T}x+b\right|}{\left\|w\right\|}$$

In order to find the optimal plane, the sum of the distances from each support vector to the hyperplane needs to be minimum, so it is only necessary to maximize ${\left\|w\right\|}^{-1}$.

In [136], the author used SVM to achieve a classification accuracy of 93.75% on the Berlin Emotion speech dataset. In [137], the author used the SVM model trained by the LDC dataset and the Emo-DB dataset to achieve an accuracy rate of 83.1% in SER based on seven emotions. In [76], the author used SVM to classify EEG signals, and achieved 85% classification accuracy. In [138], the author used SVM to perform FER on the CK+ dataset and reached 91.95% accuracy. In [139], the author used binary-SVM to realize text sentiment classification based on 15 types of emotions, and the F-score was as high as 68.86%.

#### 5.1.2. GMM

GMM aims to classify data by superimposing Gaussian distributions in a linear combination and formalize them into a probability model, as shown in Figure 12.

GMM is an unsupervised learning method, which usually uses expectation maximization (EM) [140] to determine the parameters of GMM, the main processes of EM are:Expectation: Infer optimal latent variables from the training set;Maximization: Use maximum likelihood estimation of parameters based on observed variables. It can obtain a mixture model of probabilities of all sub-distribution contained in the overall distribution. In this way, a better classification effect can be achieved without pre-determining the label of the data.

In [141], the authors achieved 82.5% accuracy on mixed gender SER by SVM based on GMM super vectors. In [142], the author used the GMM-DNN model to achieve a classification accuracy of 83.97% for six emotions. In [143], the author proposed a GMM-based federated learning framework and fully considered the privacy issues in face monitoring data, and achieved 84.1% and 74.39% accuracy for the EmotioNet and SFEW datasets.

#### 5.1.3. HMM

The Hidden Markov model (HMM) is a dynamic Bayesian network with a simple structure, which can estimate and predict unknown variables based on some observed data, as shown in Figure 13. HMM can efficiently improve the matching degree between the evaluation model and the observation sequence. HMM is able to infer hidden model states from observation sequences and better describe observed data.

The hidden variables (state variables) of the HMM can be expressed as $\left\{y_{1},y_{2},\ldots ,y_{n}\right\}$, so the state space of hidden variables contains N possible values. Observed variables can be described as $\left\{x_{1},x_{2},\ldots ,x_{n}\right\}$, and it is usually assumed that the value range of the observed variable is $\left\{o_{1},o_{2},\ldots ,o_{M}\right\}$. The system usually transitions between multiple states $\left\{s_{1},s_{2},\ldots ,s_{N}\right\}$.

The state transition probability of the model between each state is:(2)$$a_{ij}=P\left(y_{t+1}=s_{j}|y_{t}=s_{i}\right)\;1\leq i,j\leq N$$

The observed probability is:(3)$$b_{ij}=P\left(x_{t}=o_{j}|y_{t}=s_{i}\right)\;1\leq i\leq N,1\leq j\leq M$$

The initial state probability is:(4)$$\pi_{i}=P\left(y_{1}=s_{i}\right)\;1\leq i\leq N$$

At any moment, the value of the observed variable only depends on the state variable. The state variable $y_{t}$ at time t is unrelated to $y_{t-2}$ and only depends on the state variable $y_{t-1}$ at time t−1. Based on this dependence, the joint probability distribution of all variables is:(5)$$P\left(x_{1},y_{1},x_{2},y_{2},\ldots ,x_{n},y_{n}\right)=\pi_{1}b_{11}\overset{n}{\underset{i=2}{\prod }}a_{ij}b_{ij}$$

According to the above parameters, an HMM can be determined.

In [144], the authors used the HMM to classify six types of emotions for person-dependent and person-independent facial expressions, and achieved 82.46% and 58% accuracy. In [145], the authors used continuous HMMs to fully utilize low-level temporal features, and, in the SER of seven emotions, the accuracy rate was 86%. In [146], the authors developed an HMM-based audiovisual model that improved emotion recognition performance for visual and auditory signals in noisy environments. The accuracy rate of multi-modal emotion recognition in four emotions was 91.55%. In [147], the authors used the HMM for hidden sentiment detection in continuous text, and achieved 61.83% ACC and 66% AP.

#### 5.1.4. RF

Random forest (RF) [148] is a type of parallel ensemble learning. RF uses the decision tree as the base learner to construct Bagging, and further introduces random attribute selection in the training process of the decision tree. RF has a simple structure and a small amount of calculation. It can be used for both classification and regression problems. Even if the dataset is not complete, RF can maintain high classification accuracy. The increase in the classification tree does not affect the generalization performance of the classifier.

RF is an extended variant of the Bagging algorithm, as shown in Figure 14.

It randomly selects a subset containing k attributes from each node attribute set of the base decision tree, and then selects an optimal attribute from this subset for division. The parameter k controls the degree of randomness introduced. Finally, the base learners that have been trained are combined, and the majority voting method is usually used for classification prediction tasks.

In [149], the authors proposed two-layer fuzzy multiple random forest and achieved SER accuracy rates of 81.75% and 77.94% in CASIA and EmoDB datasets. In [78], the authors used RF to classify five emotions represented by HR and GSR physiological signals with an accuracy of 74%. In [150], the authors utilized multi-modal physiological signals and RF for anxiety state assessment. The classification accuracy for the five anxiety intensities reached 80.83%.

### 5.2. Deep Learning Methods

Compared to traditional machine learning methods, deep learning methods combine a feature extraction step and a classification step. With the support of large datasets, deep learning methods can learn higher-level semantic features. They have better discrimination ability for different emotions. Moreover, their generalization ability is stronger.

#### 5.2.1. CNN

As a typical deep neural network, the convolutional neural network (CNN) plays an important role in the field of emotion recognition. The convolutional neural network is mainly composed of convolutional layers, a pooling layer, a fully connected layer, and a classification layer. The convolutional layer acts as a filter to extract features of the input signal. The introduction of nonlinear factors through activation functions can enhance the expressive ability of the model. The number of parameters and calculation consumption are reduced through the pooling layer. Finally, the classification layer is used to complete the classification of the input data.

One of the earliest convolutional neural networks [151] is shown in Figure 15 (adapted from [151]). The convolutional neural network has the characteristics of parameter sharing and local connection, which makes the training of the model more efficient.

In [152], the authors built Att-Net based on CNN, and the average recall of SER in three datasets was 78.01%, 80%, and 93%. In [153], the authors proposed a CNN-RNN-based approach for dimensional emotion recognition; in FER on the gradient emotion dataset, the average concordance correlation coefficient (CCC) of the valence dimension and the arousal dimension reached up to 0.450. In [154], the authors proposed the DCNN method and achieved the best accuracies of 87.31%, 75.34%, 79.25%, and 44.61% on four FER datasets. In [155], the authors proposed a dynamical graph convolutional neural network (DGCNN) for emotion recognition on multi-channel EEG signals; the average accuracy rate in the SEED dataset and the DREAMER dataset was 90.4%. In [156], the authors proposed a 3D-CNN network framework for multimodal emotion recognition from EEG signals and face video data; the accuracy of valence dimension and arousal dimension was 96.13% and 96.79%.

#### 5.2.2. LSTM

Long Short-Term Memory (LSTM) [157] is an excellent recurrent neural network that can learn long-term dependencies from input data. At the same time, it can overcome problems such as exploding gradients and vanishing gradients. The classic LSTM framework is shown in Figure 16, which mainly includes three kinds of gate units: Input gate $i_{t}$, output gate $o_{t}$, and forget gate $f_{t}$. These gate units are used to control the information transfer of hidden state $h_{t}$, candidate state $c_{t}$, and candidate internal state ${\tilde{c}}_{t}$.

Each control gate and control state are calculated by the following formula:(6)$$i_{t}=\sigma \left(W_{i}x_{t}+U_{i}h_{t-1}+b_{i}\right)$$
(7)$$f_{t}=\sigma \left(W_{f}x_{t}+U_{f}h_{t-1}+b_{f}\right)$$
(8)$${\tilde{c}}_{t}=f\left(W_{ct}x_{t}+W_{ch}h_{t-1}+b_{c}\right)$$
(9)$$o_{t}=\sigma \left(W_{o}x_{t}+W_{o}h_{t-1}+W_{oc}c_{t}+b_{o}\right)$$
(10)$$c_{t}=f_{t}⊙c_{t-1}+i_{t}⊙{\tilde{c}}_{t}$$
(11)$$h_{t}=o_{t}⊙f(c_{t})$$

Among them, W and U represent the weight, b is the bias, the matrix σ represents the logistic function, f is the activation function, and ⊙ represents the product of vector elements.

In [158], the authors used LSTM to achieve a FER accuracy of 73.5% for six emotions based on MFCC and spectrograms features. In [159], the authors proposed a CNN-LSTM model for emotion recognition based on EEG signals. For RAW data and STD data, the accuracy rates were 90.12% and 94.17%, and the loss rates were 30.12% and 42.43%. In [160], the authors proposed the Bi-direction Long-Short Term Memory with Direction Self-Attention (BLSTM-DSA) model for SER. For the IEMOCAP dataset and the EMO-DB dataset, the overall accuracy rates were 61.20% and 85.95%, and the average accuracy rates for each category were 54.99% and 82.06%.

#### 5.2.3. DBN

The Deep Belief Network (DBN) generally consists of multiple restricted Boltzmann machines (RBM), as shown in Figure 17. RBM can avoid falling into local optimum. Each layer of the RBM is updated based on the previous layer. A DBN uses unsupervised learning and joint probability distributions to produce outputs. Hidden layer units are used to extract the correlation of high-order data in the display layer. The training of DBN mainly includes pre-training and fine-tuning.

In [161], the authors demonstrated the effectiveness of DBN in multimodal emotion recognition and achieved the best classification accuracy of 73.78% on the IEMOCAP audio-visual dataset. In [162], the authors used DBN to extract deep features from EDA, PPG, and EMG signals and then classified them. The final overall accuracy rate was 89.53%. In [163], the authors proposed a bimodal deep belief network (BDBN) to fuse speech features and expression features for multimodal emotion recognition. The classification accuracy rate on the Friends dataset was 90.89%. In [164], the author proposed a method combining PCA, LDA, and DBN; the average recognition rate in the self-built facial expression recognition dataset was 92.50%.

#### 5.2.4. Other Classification Methods

With the advancement in hardware and the improvement of computer processing power, many modern models have been proposed. They tend to have stronger classification performance and more complex structures. In order to make the classification method described in this article more comprehensive, we collected some excellent classification methods on large-scale datasets (including SER dataset, FER dataset, physiological signal dataset, and multimodal dataset). The classification method and details used by these articles are shown in Table 2 (details adapted from the cited article). Additionally, these datasets are introduced in Section 6 of this paper.

## 6. Datasets

Datasets play an important role in data-driven learning [177], which can improve the performance and robustness of models. Emotion recognition datasets are based on signal categories. According to the different signal categories, the emotion recognition datasets can be divided into: speech (textual, audio) datasets, visual (facial expression picture or video) datasets, physiological datasets, and multi-modal signal datasets.

Speech datasets for emotion recognition can be divided into performer-based [178], induced [179], and natural [180] datasets according to the method of acquisition. The performer-based datasets mainly consist of speech recordings of various emotions performed by performers with extensive experience [49]. Induced datasets are the emotions expressed by people in artificially created environments [181]. Induced datasets are relatively less expressive, but closer to reality. Natural datasets are the most realistic, usually taken from public conversations [182] or call center conversations [183], these data contain more emotional changes and background noise, but the amount is relatively limited. The commonly used speech emotion recognition datasets are shown in Table 3 (details adapted from the cited article).

For facial expression datasets, different datasets vary in terms of the acquisition environment, the number of emotion categories, age, race, image quality, etc. [196]. The commonly used facial expression recognition datasets are shown in Table 4 (details adapted from the cited article).

Physiological signals can represent more real emotions and will not be affected by people’s hidden emotional behavior. Common datasets based on physiological signals are shown in Table 5 (details adapted from the cited article).

The radar sensor can be used to obtain people’s heartbeat signals or breathing signals without contact. These sensors mainly include continuous wave radar [47,211], continuous frequency modulated wave (FMCW) radar [212], millimeter wave radar [213], and RFID tag [34]. Datasets based on radar sensor signals are less widely used according to our survey, and most researchers tend to make their own datasets. Most radar data use clipped videos or pictures as emotional inducers. Radar sensors are used to collect physiological signals of volunteers to make datasets.

Commonly used multi-modal emotion recognition datasets are shown in Table 6 (details adapted from the cited article). The multimodal signal dataset contains at least two different signals and richer information. Multi-modal emotion recognition datasets often require a larger amount of data, and the data usually needs to be labeled. Therefore, making multimodal signal datasets becomes more difficult than normal datasets.

At the same time, we also need to consider the synchronization of multi-channel signals during the recording of different sensors, as some devices may record on different time scales. Multi-modal signal datasets can make the machine’s analysis of emotion more comprehensive. At present, researchers are paying increasing attention to multimodal emotion recognition.

## 7. Conclusions and Discussions

In this survey, we reviewed more than 200 papers, including working processes, methods, and commonly used datasets of different sensors for emotion recognition. In this section, we summarize the main findings from this survey.

Facial expressions can intuitively reflect the subjective emotions in interpersonal communication, but they are affected by limited lighting, occlusion, small changes in facial expressions, and individual differences. The performances of existing vision-based emotion recognition systems will significantly drop in environments with changing lighting conditions. Self-occlusion due to head rotation or face contact, and occlusion by other people passing in front of the camera, are both common problems. Moreover, individual differences can affect the feature extraction and learning of the model. There are large differences between infants and adults, males and females, and different groups, which makes it challenging to train a FER classifier with strong generalization performance.

SER is also of great significance in emotion recognition. Due to the variability of emotions, a piece of speech often has multiple emotions, which is challenging for the accurate extraction of speech information features. For multiple languages, cross-cultural emotion recognition is the future development trend. People in different countries and regions have certain cultural differences, but, for humans, even if they cannot understand what foreigners are saying, they can roughly understand their tone and attitude.

Emotional changes are also reflected in the physiological changes of the human body. The most basic challenge of emotion recognition from physiological signals is the accurate emotional labeling of data. In real life, parties often do not realize that they have developed certain emotions, because the parties are caught in the emotion itself. Therefore, participants need to exactly record when a certain emotion occurs. Only in this way can the corresponding physiological signals be extracted. Some physiological signal recording devices are expensive and invasive, which greatly limits the number of subjects and the length of the experiment. Therefore, some non-contact physiological signal recording devices are more popular.

Multi-modal emotion recognition based on multi-sensors can make up for the deficiency of single sensor. It is more robust and is now receiving more attention. It uses different signals to extract features and perform feature-level or decision-level fusion to a certain extent, which can improve the accuracy of discrete or dimensional sentiment classification. The main challenges include how to choose an appropriate feature representation method and feature selection method based on multi-modal signal input. Different modal signals may also have mutual dependencies in different time dimensions, and classifiers need to be designed according to the potential correlation of different modal datasets.

We introduced several commonly used classical machine learning and deep learning classification methods. Classical machine learning methods have faster speeds and simpler structures. However, for large and high-dimensional data, researchers prefer deep learning. With the improvement in computer technology, deeper and larger deep learning models have been proposed, which can extract high-dimensional features better. However, this does not mean that classic machine learning methods are abandoned. For limited training data, machine learning often achieves better results.

Based on the above conclusions, we think that single-modal emotion recognition cannot meet human needs in some specific application scenarios. Therefore, the current research on multimodal information processing is more popular. However, the research on multimodal emotion recognition has more challenges. They include experimental environment, sensors, signal acquisition, signal processing, information annotation, etc. At the same time, we believe that emotion recognition is an important part of the development of artificial intelligence. Accurate recognition of emotions can enable machines to better serve people and care about people’s health and life in more detail.

## 8. Future Trends

Emotion recognition is of great significance to both human and social development. The current challenges and development trends of emotion recognition mainly include technical aspects and security aspects.

The first is to improve user acceptance. At present, many people are not familiar with various emotional computing sensors, and some sensors need to be worn by users. In order to improve the degree of cooperation of users, practitioners need to give detailed instructions to users. The detection system should also be user-centered, with the primary goal of protecting the user’s physical and mental health.

The second point is security. The process of human emotion recognition involves highly private personal information, including health, location, and physiological characteristics. Emotion recognition should be used in socially beneficial research rather than being used to cause legal problems or discrimination. Therefore, protecting user privacy has also become a major challenge for emotion recognition. At present, decentralized AI technology can overcome the limitations of centralized information storage and improve data privacy and security.

The third point is robustness and accuracy. The current emotion recognition model cannot simulate all aspects of human emotions. In order to be more comprehensive, multimodal emotion recognition has become the first choice for most researchers. With larger models and datasets, multimodal approaches can achieve better results. Emotion recognition often requires more information, and short-term or transient features can only represent people’s psychological state at a specific time. Studies on personality analysis, such as autism diagnosis and intelligence testing, require longer-term observation. Therefore, the extraction of long-term features is also challenging and of great research significance for emotion recognition.

In order to obtain a good emotion recognition model, there are more stringent requirements for datasets. With the continuous production of large-scale datasets, the advantages of unsupervised learning and reinforcement learning are more obvious. Unsupervised learning does not require pre-stored labels or specifications. Moreover, it can also complete classification without category information. Reinforcement learning enables the model to maximize rewards through the principle of trial and error, and can continuously optimize the performance of the system. These emotion recognition methods are also worthy of research.

## Figures and Tables

**Figure 1 sensors-23-02455-f001:**
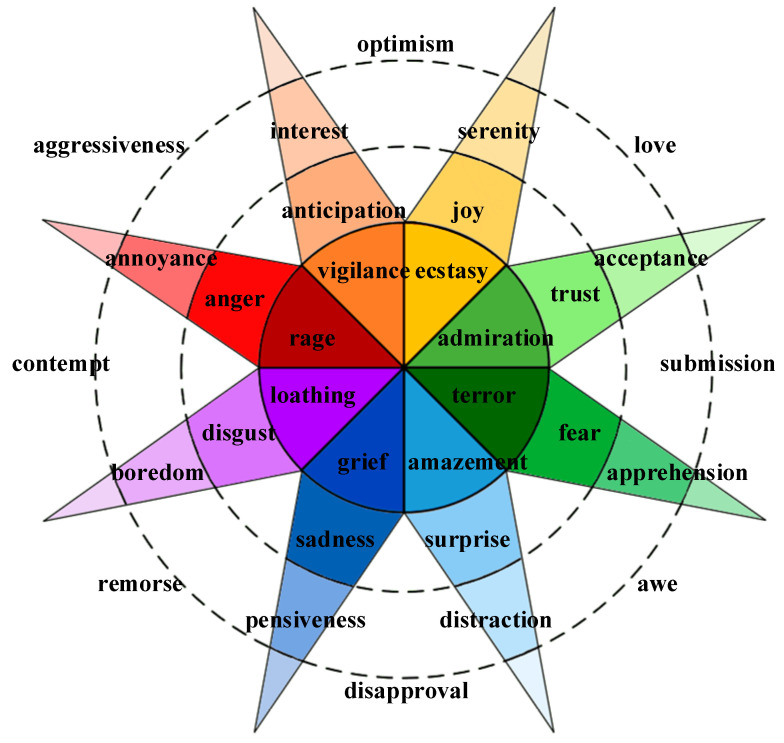
Piutchik’s wheel model.2.2. Dimensional Emotion Model.

**Figure 2 sensors-23-02455-f002:**
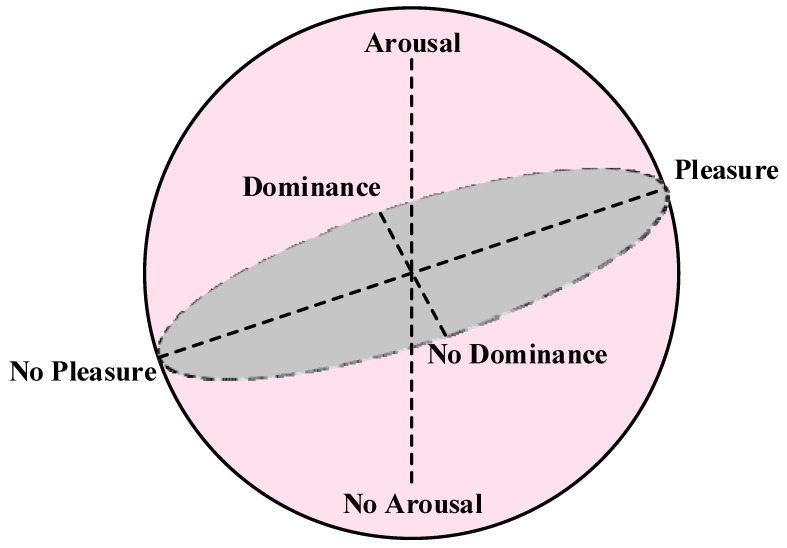
PAD 3D emotion model.

**Figure 3 sensors-23-02455-f003:**
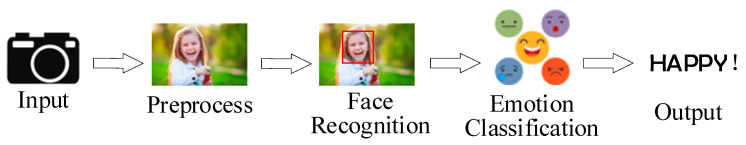
Facial expression recognition process.

**Figure 4 sensors-23-02455-f004:**
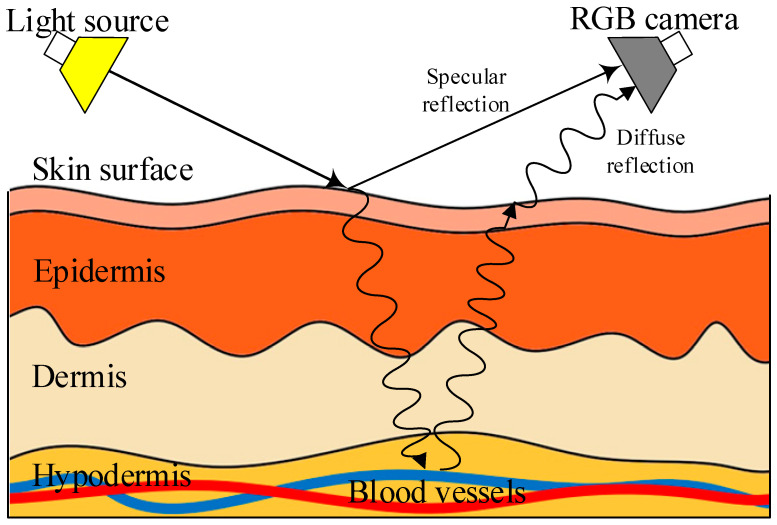
Schematic diagram of rPPG technology.

**Figure 5 sensors-23-02455-f005:**
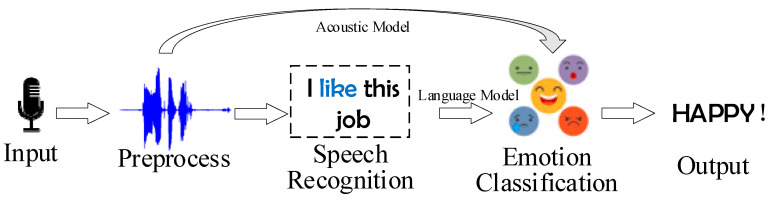
The process of SER.

**Figure 6 sensors-23-02455-f006:**
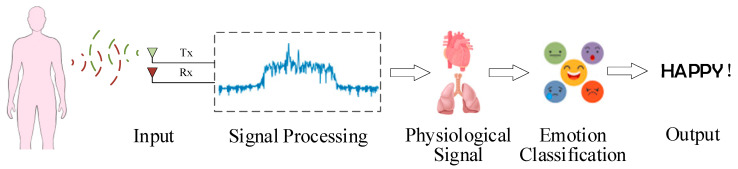
Process of emotion recognition based on radar sensor.

**Figure 7 sensors-23-02455-f007:**
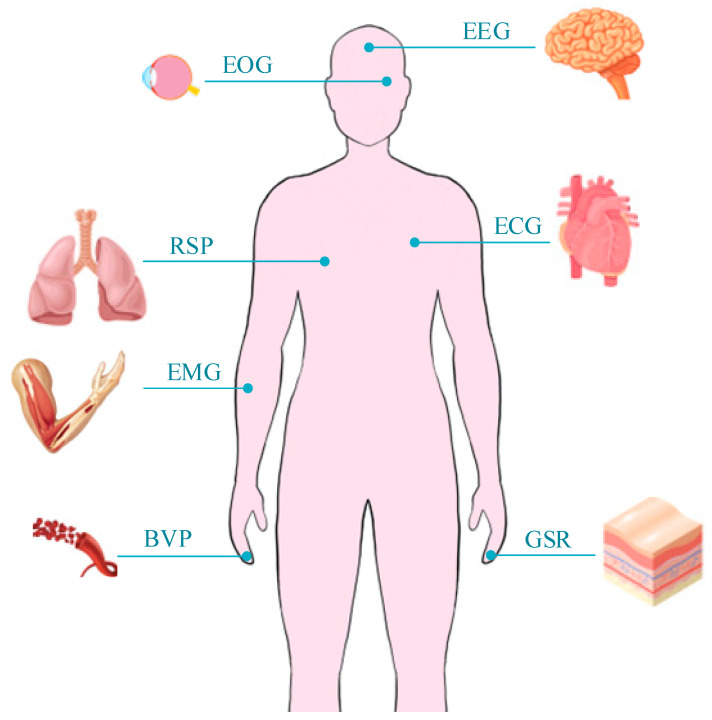
Physiological signals detected by other physiological sensors.

**Figure 8 sensors-23-02455-f008:**
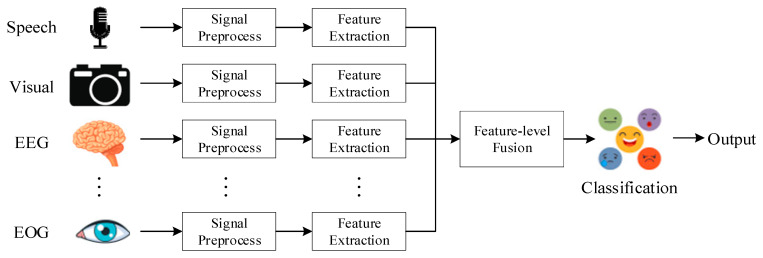
Feature-level fusion.

**Figure 9 sensors-23-02455-f009:**
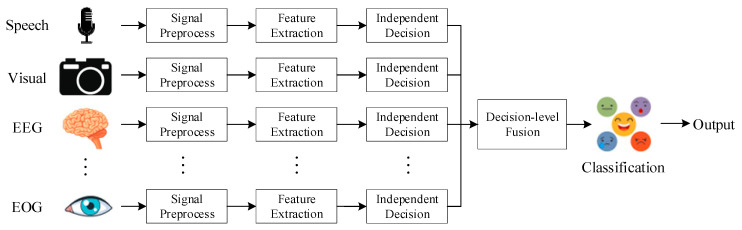
Decision-level fusion.

**Figure 10 sensors-23-02455-f010:**
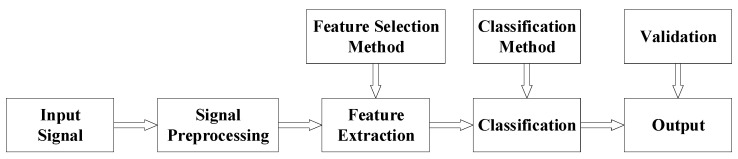
Process of emotion recognition method.

**Figure 11 sensors-23-02455-f011:**
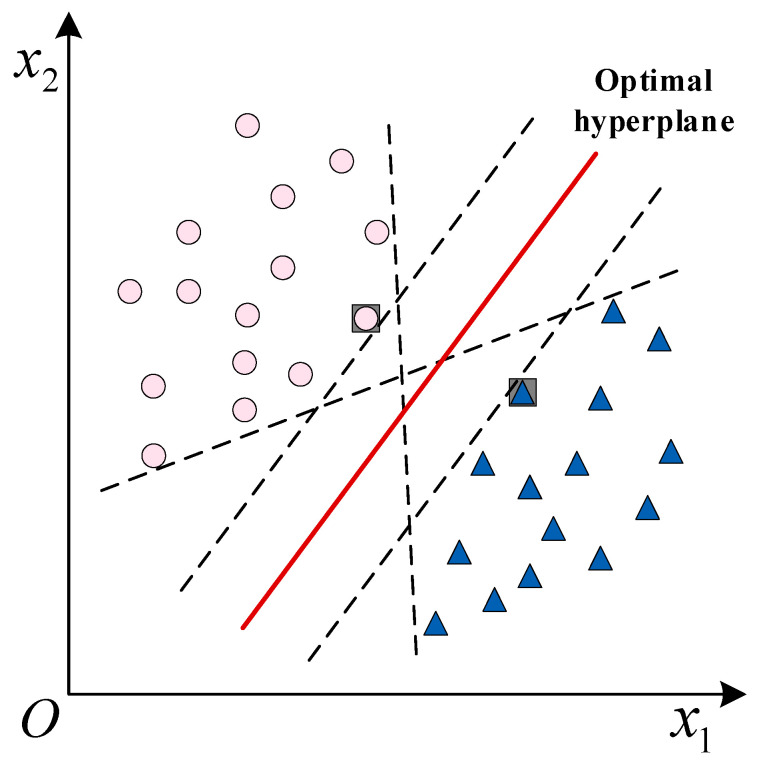
Support vectors in two-dimensional space.

**Figure 12 sensors-23-02455-f012:**
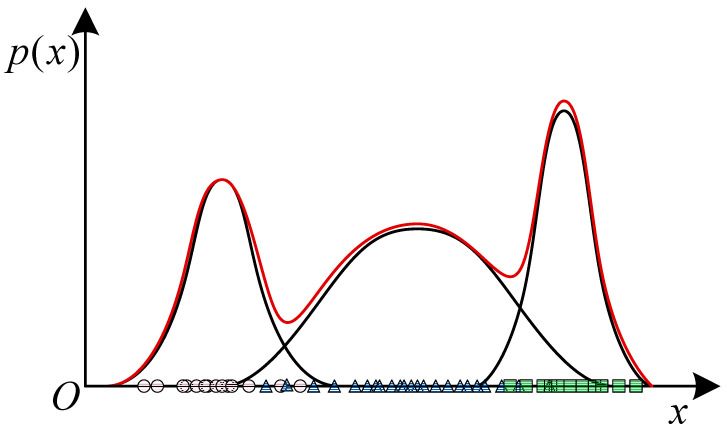
GMM classification.

**Figure 13 sensors-23-02455-f013:**
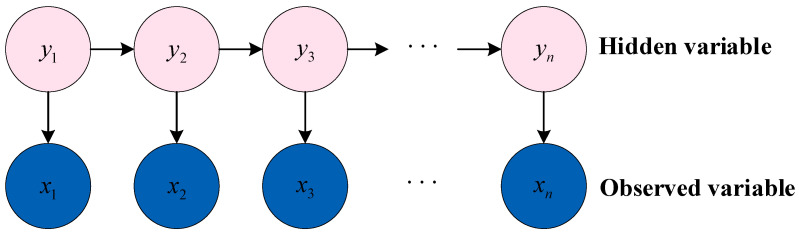
Structure of HMM.

**Figure 14 sensors-23-02455-f014:**
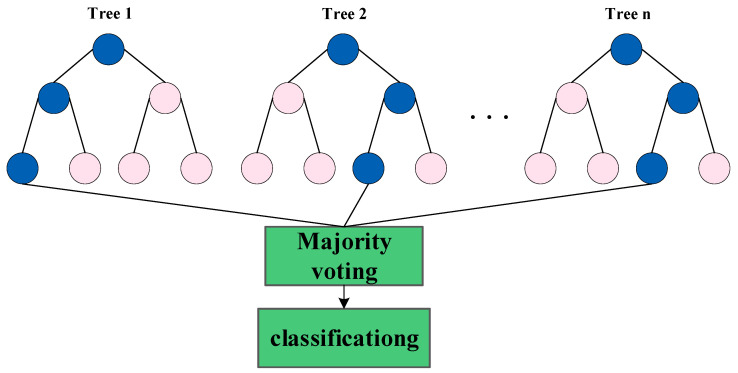
Structure of RF.

**Figure 15 sensors-23-02455-f015:**
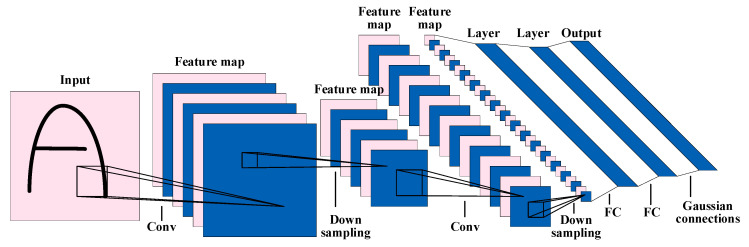
The network structure of LeNet.

**Figure 16 sensors-23-02455-f016:**
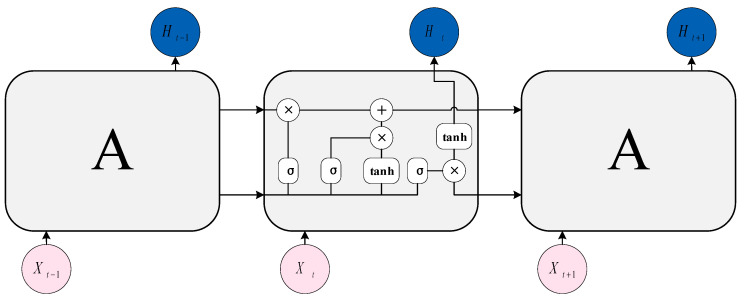
The structure of LSTM.

**Figure 17 sensors-23-02455-f017:**
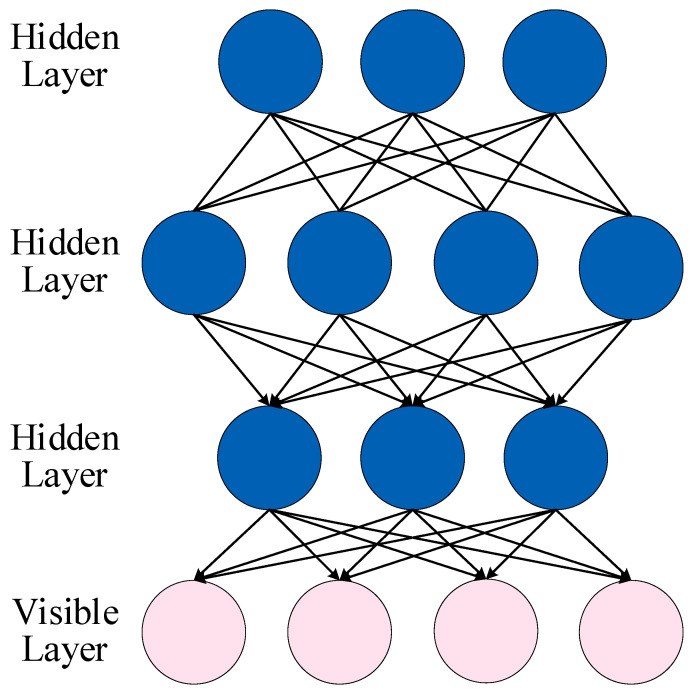
The structure of DBN.

**Table 1 sensors-23-02455-t001:** Advantages and disadvantages of different sensors for emotion recognition.

Sensors	Advantages	Disadvantages
Visual sensor	Simple data collection; high scalability	Restricted by light; easy to cause privacy leakage [26]
Audio sensor	Low cost; wide range of applications	Lack of robustness for complex sentiment analysis
Radar sensor	Remote monitoring of physiological signals	Radial movement may cause disturbance
Other physiological sensors	Ability to monitor physiological signals representing real emotion	Invasive, requires wearing close to the skin surface [27]
Multi-sensor fusion	Richer collected information; higher robustness	Multi-channel information needs to be synchronized; the follow-up calculation is relatively large

**Table 2 sensors-23-02455-t002:** Other classification methods.

Model Name	Dataset Used	Classification Method	Details
T5-3B [165]	SST (NLP)	Transformer and self-attention	The authors used transfer learning and self-attention to convert all text-based language problems into a text-to-text format. The authors compared the pre-training objectives, architectures, unlabeled datasets, and transfer methods of NLP. The classification accuracy on the SST dataset is 97.4%.
MT-DNN-SMART [166]	SST (NLP)	Transformer and smoothness inducing regularization	The authors proposed smoothness-induced regularization based on transfer learning to manage the complexity of the model. At the same time, a new optimization method was proposed to prevent over-updating. The classification accuracy on the SST dataset is 97.5%.
GRU [167]	CREMA-D (SER)	Self-supervised representation learning	The authors proposed a framework for learning audio representations guided by the visual modality in the context of audiovisual speech. The authors demonstrated the potential of visual supervision for learning audio representations; and achieved 55.01% SER accuracy on the CREMA-D dataset.
EmoAffectNet [168]	CREMA-D and AffectNet (FER)	CNN-LSTM	The authors proposed a flexible FER system using CNN and LSTM. This system consists of a backbone model and several temporal models. Every component of the system can be replaced by other models. The backbone model achieved an accuracy of 66.4% on the AffectNet dataset. The overall model achieved an accuracy of 79% on the CERMA-D dataset.
M2FNet [169]	IEMOCAP and MELD (multimodal)	Multi-task CNN and multi-head attention-based fusion	The multimodal fusion network proposed by the authors can extract emotional features from visual, audio, and textual modalities. The feature extractor was trained by an adaptive margin-based triplet loss function. The model achieved 67.85% accuracy and a 66.71 weighted average F1 score on the MELD dataset. Meanwhile, it achieved 69.69% accuracy and a 69.86 weighted average F1 score on the MELD dataset.
CH Fusion [170]	IEMOCAP (multimodal)	RNN and feature fusion strategy	The authors used RNN to extract the unimodal features of the three modalities of audio, video, and text. These unimodal features were then fused through a fully connected layer to form trimodal features. Finally, feature vectors for sentiment classification were obtained. The model achieved an F1 score of 0.768 and an accuracy rate of 0.765 on the IEMOCAP dataset.
EmotionFlow-large [171]	MELD (multimodal)	BERT model and Conditional random field (CRF)	The authors researched the propagation of emotions in dialogue emotion recognition. The authors utilized an encoder-decoder structure to learn user-specific features. Conditional random fields (CRF) were then applied to capture sequence information at the sentiment level. The weighted F1 score on the MELD dataset was 66.50.
FN2EN [172]	CK+ (FER)	DCNN	The authors proposed a two-stage training algorithm. In the first stage, high-level neuronal responses were modeled using probability distribution functions based on the fine-tuned face network. In the second stage, the authors conducted label supervision to improve the discriminative ability. The model achieved 96.8% (eight emotions) and 98.6% (six emotions) accuracy on the CK+ dataset.
Multi-task EfficientNet-B2 [173]	AffectNet (FER)	MTCNN and Adam optimization	In the article, the authors analyzed the behavior of students in the e-learning environment. The facial features obtained by the model could be used to quickly predict student engagement, individual emotions, and group-level influence. The model could even be used for real-time video processing on each student’s mobile device without sending the video to a remote server or the teacher’s PC. The model achieved 63.03% (eight emotions) and 66.29% (seven emotions) accuracy on the AffectNet dataset.
EAC [174]	RAF-DB (FER)	CNN and Class Activation Mapping (CAM)	The authors approached noisy label FER from the perspective of feature learning, and proposed Erase Attention Consistency (EAC). EAC does not require noise rate or label integration. It can generalize better to noisy label classification tasks with a large number of classes. The overall accuracy on the RAF-DB dataset was 90.35%.
BiHDM [175]	SEED (EEG signal)	RNNs	The authors proposed a model to learn the differential information of the left and right hemispheres of the human brain to improve EEG emotion recognition. The authors employed four directed recurrent neural networks based on two orientations to traverse electrode signals on two separate brain regions. This preserved its inherent spatial dependence. The accuracy on the SEED dataset reached 74.35%.
MMLatch [176]	CMU-MOSEI (multimodal)	LSTM, RNNs and Transformers	The neural architecture proposed by the authors could capture top-down cross-modal interactions. A forward propagation feedback mechanism was used during model training. The accuracy rate on the CMU-MOSEI dataset reached 82.4.

**Table 3 sensors-23-02455-t003:** Dataset for speech emotion recognition.

Name	Type	Details	Number of Emotion Categories	Number of Samples
MDS [184]	Textual	Product reviews from the Amazon shopping site; consisting of different words, sentences, and documents	2 or 5	100,000
SST [185]	Textual	Semantic emotion recognition database established by Stanford University	2 or 5	11,855
IMDB [186]	Textual	Contains a large number of movie reviews	2	25,000
EMODB [187]	Performer-based	The dataset consists of ten German voices spoken by ten speakers (five males and five females)	7	800
SAVEE [188]	Performer-based	Performed by four female speakers; spoken in English	7	480
CREAM-D [189]	Performer-based	Spoken in English	6	7442
IEMOCAP * [190]	Performer-based	Conversation between two people (one male and one female); spoken in English	4	-
Chinese Emotion Speech Dataset [191]	Induced	Spoken in Chinese	5	3649
MELD * [192]	Induced	Data from TC-series Friends	3	13,000
RECOLA Speech Database [179]	Natural	Spoken by 46 speakers (19 male and 27 female); spoken in French	5	7 h
FAU Aibo emotion corpus [193]	Natural	Communications between 51 children and a robot dog; spoken in German	11	9 h
Semaine Database [194]	Natural	Spoken by 150 speakers; spoken in English, Greek, and Hebrew	5	959 conversations
CHEAVD [195]	Natural	Spoken by 238 speakers (from children to the elderly); spoken in Chinese	26	2322

* Can also be used for multimodal emotion recognition.

**Table 4 sensors-23-02455-t004:** Datasets for facial expression recognition.

Name	Type	Details	Number of Emotion Categories	Number of Samples
BP4D [197]	Induced	41participants; 4 ethnicities; 18–29 years old	8	368,036
CK+ [198]	Induced	123 participants; 23 facial displays; 21–53 years old	7	593 sequences
BU-4DEF [199]	Induced	101 participants; 5 ethnicities	6	606 sequences
SEWA [200]	Induced	96 participants; 6 ethnicities; 18–65 years old	7	1990 sequences
MMI-V [201]	Performer-based	25 participants; 3 ethnicities; 19–62 years old	6	1.5 h
JAFFE [202]	Performer-based	10 participants	6	213
BU-3DEF [203]	Performer-based	100 participants 18–70 years old	6	2500
AffectNet [204]	Natural	Average age is 33.01 years old; downloaded from the Internet	6	450,000
RAF-DB [205]	Natural	Collected from Flickr	compound	29,672
EmotioNet [206]	Natural	Downloaded from the Internet	compound	1,000,000

**Table 5 sensors-23-02455-t005:** Datasets of physiological signals.

Name	Type	Details	Number of Emotion Categories	Physiological Signals
DEAP * [207]	Induced	32 participants; average age is 26.9 years old	Dimensional emotion (arousal-valence-dominance)	EEG; EMG; RSP; GSR; EOG; plethysmograph; skin temperature
DECAF * [208]	Induced	30 participants	Dimensional emotion (arousal-valence)	EMG; NIR; hEOG; ECG; tEMG
AMIGOS * [209]	Induced	Individual participant and group participants	Dimensional emotion (arousal-valance)	EEG; GSR; ECG
SEED * [210]	Induced	15 participants; average age is 23.3	3	EEG; EOG
DREAMER * [77]	Induced	23 participants; collected by wireless low-cost off-the-self devices	Dimensional emotion (arousal-valance-dominance)	EEG; ECG

* Can also be used for multimodal emotion recognition.

**Table 6 sensors-23-02455-t006:** Datasets for multi-modal emotion recognition.

Name	Type	Details	Number of Emotion Categories	Types of Signals
eNTERFACE [49]	Induced	42 participants; 14 different nationalities	6	Visual signals; audio signals;
RECOLA [179]	Natural	46 participants; 9.5 h	Dimensional emotion (arousal-valence)	Visual signals; audio signals; ECG signals; EDA signals
CMU-MOSEI [214]	Natural	23,453 annotated video segments; 1000 speaker; 250 topics	6	Textual signals visual signals; audio signals;
MAHNOB-HCI [215]	Induced	27 participants	Dimensional emotion (arousal-valence-dominance)	Textual signals visual signals; audio signals; EEG signals; RSP signals; GSR signals; ECG signals; skin temperature signals

## Data Availability

Some of the datasets mentioned in the paper can be downloaded in: https://paperswithcode.com.

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
