# Peer review of "Emotion Recognition Using Different Sensors, Emotion Models, Methods and Datasets: A Comprehensive Review"

_sensors, 2023, doi:10.3390/s23052455_

Round 1

Reviewer 1 Report

The paper propose a comprehansive survey related emotion recognition using the several modalities. Authors say about visual modality, audio modality, and sensor that can monitor human. In general the paper is interesting. Authors summarise main trends in the topic done last 10 years. However, for me it is not clear how emations are related to the human chracteristics: blood pressure, HR, and etc. It shouild be clarified in details. Also I propose to enhance the absract and introduction, say more about the survay and scientific novelty.

Author Response

Thank you for your valuable comments and suggestions. Please see the attachment.

Reviewer 2 Report

This work presents a review of systems for emotion recognition. However, the paper does not present anything new. The authors have gathered articles related to the topic, but did not conduct relevant analyses related to emotion recognition. Additionally, the criteria for selecting articles, the review methodology, and aspects related to emotion recognition work were not presented.

The English needs to be reviewed. Some sentences are too long, and need to be split.

Major Issues:

-The information regarding the method used during the review and the results obtained, even qualitatively, is lacking. Furthermore, the abstract lacks a more comprehensive conclusion.

-The information is not in the correct order. For example, section 3 starts talking about emotion models, while section 2 already dealt with emotion detection systems. I suggest reviewing these parts of the text to avoid a break in the subject during reading.

-In section 4.2, the authors could have deepened their analysis of the types of features that are most commonly applied for each type of signal, their impacts, a quantitative evaluation, and the proportion of articles that use certain techniques, as well as their success rates. The data provided by the authors is vague and only serves to illustrate the topic, but does not have the depth that a review article requires.

-Section 5 presents details on how classification techniques work, which can be found in any book or material on the subject. The part related to its use in emotion recognition is very superficial.

-A discussion section is missing.

Minor Issues:

-Verify the use of commas in enumeration sentences.

-Use "Sections" instead of "Chapter" throughout the text.

-The authors should follow the template for referencing figures and tables.

-Improve the quality of the figures.

-Review the end of sentence 105.

-In Figure 8, use a reference such as "adapted from (...)."

-Lines 388-389 are broken.

-Some references are preprints and have not gone through peer review. Only peer-reviewed references should be used in scientific research

Author Response

(The authors gave the same response as above.)

Reviewer 3 Report

The article focuses on various aspects of deep machine learning technologies for automatic emotion recognition. It serves as a comprehensive review that provides in-depth and clear explanations of the key definitions related to emotion recognition using various information capture devices. The article also includes numerous informative figures and comparative tables, which is a great advantage. Overall, the article is easily readable and the material presented by the authors is comprehensible. However, in my opinion, there are a few shortcomings that need to be addressed to improve the quality of the article.

1) First and foremost, it is notable that the authors discuss emotion classification methods such as SVM, GMM, HMM, RF, CNN, LSTM, and DBN, which is commendable. However, it is surprising that there is no mention of more recent, both unimodal and multimodal, emotion recognition methods. Some of the classifiers described, such as SVM, GMM, HMM, and RF, are no longer widely used. In my opinion, the authors should expand on the description of modern works in this field. For example, if we take the very famous EmoAffectNet corpus (which the authors have in the table), then paperswithcode has a rating table for the best works (https://paperswithcode.com/sota/facial-expression-recognition-on-affectnet). All the best works there are modern. Therefore, it is recommended to add a description, for example, 3 best results for recognizing 7 basic emotions by video modality (Emotion-GCN - https://paperswithcode.com/paper/exploiting-emotional-dependencies-with-graph, EmoAffectNet - https://paperswithcode.com /paper/in-search-of-a-robust-facial-expressions, Multi-task EfficientNet-B2 - https://paperswithcode.com/paper/classifying-emotions-and-engagement-in-online) to the description section videomodality. The same can be done for audio and multimodal recognition (the links provided for an example of work are also in other ratings on paperwithcode). At the discretion of the author, one can also take lesser known datasets and select better results from them. In addition, adding such a description will expand links to previous works of the global scientific community (2020-22), which are constantly presented at conferences focused on working with audio or video modalities (CVPR, ICCV, INTERSPEECH, ICASSP, EUSIPCO, ICMI and others) or in Q1 journals.

2) If possible, present the drawings in vector format, this will further improve the article visually.

3) The abstract can be expanded, it is understandable, but too short.

4) Keywords can also be expanded.

5) The style of the article requires minor revision due to the presence of spelling and punctuation errors, but even now the article is easy to read and everything is very clear.

At this stage, there are shortcomings in the review article that should be eliminated. However, it is worth noting that in general the article deserves attention, but it is better to finalize it.

Author Response

(The authors gave the same response as above.)

Round 2

Reviewer 2 Report

The authors performed the requested reviews. It is need to complete the author's contribution and acknowledgements.

Reviewer 3 Report

The authors of the article have made substantial improvements to the manuscript, addressing all comments and correcting identified shortcomings. With all sections now completed, the article should prove valuable and interesting to many specialists whose research relates to deep machine learning technologies for automatic emotion recognition. Therefore, the article is highly suitable for publication.